# Targeted Selected Reaction Monitoring Verifies Histology Specific Peptide Signatures in Epithelial Ovarian Cancer

**DOI:** 10.3390/cancers13225713

**Published:** 2021-11-15

**Authors:** Leena Liljedahl, Johan Malmström, Björg Kristjansdottir, Sofia Waldemarson, Karin Sundfeldt

**Affiliations:** 1CREATE Health Translational Cancer Center, Department of Immunotechnology, Lund University, Medicon Village, 223 81 Lund, Sweden; leena.liljedahl@immun.lth.se (L.L.); sofia.waldemarsson@immun.lth.se (S.W.); 2Department of Clinical Sciences, Lund University, BMC D13, 221 84 Lund, Sweden; johan.malmstrom@med.lu.se; 3Sahlgrenska Center for Cancer Research, Department of Obstetrics and Gynecology, Institute of Clinical Sciences, University of Gothenburg, 405 30 Gothenburg, Sweden; Bjorg.Kristjansdottir@vgregion.se

**Keywords:** epithelial ovarian cancer (OC), targeted selected reaction monitoring (SRM), proteomics, biomarkers, early-stage diagnostics

## Abstract

**Simple Summary:**

Ovarian cancer is a lethal disease due to its late phase discovery. Any steps towards improving early diagnostics will dramatically increase survival rates. To identify new ovarian cancer biomarker panels, we need to focus on early-stage disease and all histologic subtypes. In this study we have, based on prior discoveries, constructed a multiplexed targeted selected-reaction-monitoring assay to detect peptides from 177 proteins in only 20 µL of plasma. The assay was evaluated in patients with a focus on early-stages and all ovarian cancer histologies in separate groups. With multivariate analysis, we found the highest predictive value in the benign vs. low-grade serous (Q2 = 0.615) and mucinous (Q2 = 0.611) early stage compared to all malignant (Q2 = 0.226) or late stage (Q2 = 0.43) ovarian cancers. The results show that each ovarian cancer histology subgroup can be identified by a unique panel of proteins.

**Abstract:**

Epithelial ovarian cancer (OC) is a disease with high mortality due to vague early clinical symptoms. Benign ovarian cysts are common and accurate diagnosis remains a challenge because of the molecular heterogeneity of OC. We set out to investigate whether the disease diversity seen in ovarian cyst fluids and tumor tissue could be detected in plasma. Using existing mass spectrometry (MS)-based proteomics data, we constructed a selected reaction monitoring (SRM) assay targeting peptides from 177 cancer-related and classical proteins associated with OC. Plasma from benign, borderline, and malignant ovarian tumors were used to verify expression (*n* = 74). Unsupervised and supervised multivariate analyses were used for comparisons. The peptide signatures revealed by the supervised multivariate analysis contained 55 to 77 peptides each. The predictive (Q2) values were higher for benign vs. low-grade serous Q2 = 0.615, mucinous Q2 = 0.611, endometrioid Q2 = 0.428 and high-grade serous Q2 = 0.375 (stage I–II Q2 = 0.515; stage III Q2 = 0.43) OC compared to benign vs. all malignant Q2 = 0.226. With targeted SRM MS we constructed a multiplexed assay for simultaneous detection and relative quantification of 185 peptides from 177 proteins in only 20 µL of plasma. With the approach of histology-specific peptide patterns, derived from pre-selected proteins, we may be able to detect not only high-grade serous OC but also the less common OC subtypes.

## 1. Introduction

High-grade serous ovarian cancer (HGSC) accounts for 65–70% of all ovarian cancers (OC) and, consequently, are the ones that define the poor prognosis of the disease [1]. The earlier cancer can be detected, the better the chance for a successful treatment. OC’s 5-year survival increases from approximately 30% to 90% if diagnosed in early stages. The localized cancer can be treated with surgery alone, leaving adjuvant chemotherapy superfluous. OC is a highly diverse disease with several histologic subtypes. Supported by new insights in molecular genetics, OC was nearly 20 years ago designated two major groups, Type 1 and Type 2, revised in 2014 [2]. The Type 1, which included low-grade serous carcinoma (LGSC), endometrioid, clear-cell and mucinous OC, is in fact very heterogenous, while Type 2, dominated by HGSC, is more homogenous. Regardless of histological subtypes, improved diagnostics in early stages alone hold great potential to save many lives.

Independent of type, all OC require surgery but to a varying degree, based on stage and histologic subtype [3]. Difficulties in identifying the malignant tumors among the many benign ovarian cysts that present with symptoms or incidentally using various imaging technologies are ever-present in today´s clinical routines [4]. The incidence of OC is low, with a life-time risk of 1.5%, while benign cysts are common, particularly before menopause, with a prevalence of 36%. Symptoms may be absent and are vague. Patients with suspicious ovarian cysts are typically examined using transvaginal ultrasound. There are a couple of serum biomarkers that are occasionally used in OC diagnostics, serum marker cancer antigen 125 (CA125), human epididymis 4 (HE4) and algorithms based on the above: Risk of Ovarian Cancer Algorithm (ROCA), the Risk of Malignancy Algorithm (ROMA), Risk Malignancy Index (RMI) and Multivariate Index Assay (MIA) [5,6,7,8,9]. The lack of specificity and sensitivity in all early stages and in women below 50 years drives the continued development of better tools for OC diagnostics or screening. The above-mentioned biomarkers and algorithms have been discovered in mixed OC cohorts where HGSC stage III in an elderly population has been the dominating histologic tumor type. Accordingly, both CA125 and HE4 have acceptable sensitivity for ovarian cyst/pelvic tumor differential diagnostics, especially for late-stage HGSC in postmenopausal women.

Improved molecular classification of OC provides better opportunities for biomarker discovery comparing more relevant histologic subtypes in the search for novel biomarkers. Our hypothesis is that each tumor group can be identified by a subset of proteins specific for each OC histology subtype. In earlier studies, we explored ovarian tissue and early-stage ovarian cyst fluid for the discovery of biomarker candidates [10,11,12,13,14]. The findings from these studies, together with selected literature references [15,16,17,18] were the basis for the current study. Candidate proteins were selected, and peptides were tested for analysis in a mixed cohort of OC histologies primarily in early stages. We applied a state-of-the-art liquid chromatography tandem mass spectrometry (LC-MS/MS) workflow to screen for these markers in the plasma samples followed by the development of a multiplexed targeted selected reaction monitoring (SRM) assay for the candidates.

## 2. Materials and Methods

### 2.1. Study Design—Workflow

Phase I—Discovery—Biomarker discovery was performed in OC tumor biopsies and ovarian cyst fluids using shotgun mass spectrometry (Figure 1) [10,12,13]. Data from a total of 9 studies, our own previous work and selected literature references, were compiled to create a biomarker target list of 284 proteins. Phase II—Verification—Proteins differentially expressed comparing OC histology subgroups and benign subjects were evaluated in patient plasma using targeted mass spectrometry applying SRM [19]. The results from the SRM analysis were explored using state-of-the-art bioinformatic approaches.

### 2.2. Study Population

Samples were collected prior to surgery from patients presenting with a suspected malignant pelvic mass at the section for gynecology cancer surgery, Sahlgrenska University hospital, Gothenburg, between 2012–2015 (Figure 1). A total of 74 patient plasma samples were selected retrospectively from the gynecology tumor biobank (Biobank Väst). Of the 74 included samples, 16 were benign, 13 borderline type tumors, seven endometrioid OC stage I–II, four LGSC stage I–II, nine mucinous OC stage I–II, eight HGSC stage I–II, and 17 HGSC stage III (Table 1). All resected tumors were examined by an experienced reference pathologist in gynecology for diagnosis, histology, grade and staging (I–III) according to International Federation of Gynecology and Obstetrics (FIGO) standards. All patients signed a written informed consent according to the requirements set by the local ethics committee in Gothenburg, in accordance with the Declaration of Helsinki (Dnr 445–2008-08-18; 139–2013-04-25).

### 2.3. Plasma Sample Preparation

Six milliliters (mL) of blood was collected in EDTA plasma vacutainers after anesthesia but before surgery using standardized procedures and stored at 4 °C within 15–30 min. After centrifugation, plasma was collected and directly aliquoted into Eppendorf tubes, frozen and stored at −80 °C within 30–60 min after withdrawal. All patient samples included were handled and processed according to standardized procedures. Prior to analysis, the samples were thawed on ice and depleted (according to the manufacturer´s specification) using the Agilent 14 protein depletion spin cartridge (HU-14 5188–6560, Agilent, Santa Clara, CA, USA) resulting in up to 96% removal of the 14 abundant blood proteins albumin, IgG, antitrypsin, IgA, transferrin, haptoglobin, fibrinogen, alpha2-macroglobulin, alpha1-acid glycoprotein, IgM, apolipoprotein A1, apolipoprotein A2, complement C3 and transthyretin. Peptides from eleven of those (all except IgG, IgA, IgM) were included in the SRM MS analysis to verify that the depletion procedure was successful for all samples. After depletion, samples were reduced, alkylated, and digested with 10 µg/mL sequencing grade modified Trypsin (Promega, Madison, WI, USA) ON at 37 °C, then separated by RP-C18 chromatography (Waters, Milford, MA, USA), dried on a Speed-vac (Thermo Scientific, Waltham, MA, USA) and frozen in −80 °C until MS analysis.

### 2.4. Selection of Target Peptides and Transitions

A spectral library was built into the SRM management software Skyline (PMID: 20147306) from in-house over time accumulated shotgun MS-MS/MS spectra obtained from plasma analyses on a QExactive mass spectrometer (Thermo Electron, Bremen, Germany) as described in [20,21]. The list of target proteins generated from our previous studies [10,11,12,13,14] and literature [15,16,17,18,22] (see below) was imported and in-silico digested in Skyline. The in-house generated plasma library was used as the primary library for selection of peptides and transitions. Peptide quality was based on total intensity in MS1 spectrum and % of total spectrum intensity contributed by identified transitions in the MS2 spectrum. The top 3 unique peptides per protein identified with highest MS1 spectrum intensity were selected. Out of these, the 1–2 peptides with the cleanest MS2 spectrum, meaning the highest % of total MS2 spectrum intensity contributed by identified peptide fragments were selected. Any protein where two peptides met the criterium of five good transitions (y- and b-ion signals) was selected for ordering a synthetic peptide. For any protein where only one peptide met this criterum, only one peptide was selected. Peptide identity and quality and final transition selection was subsequently verified using the synthetic peptides. For proteins not represented in the plasma library, a public library of human plasma analyses was downloaded from National Institute of Standards and Technology (NIST) and peptides were selected using the above criteria. For proteins not represented in the NIST library, the SRM atlas was scanned for peptides and 2–3 representatively unique peptides per protein were considered for the assay.

### 2.5. Protein Selection

Protein candidates were selected based on candidate markers from our previous biomarker discovery studies in tissues and in ovarian cyst fluids, clinically used biomarkers CA125, HE4 and the MIA panel, biomarker candidates from the literature and unpublished data, prioritizing markers that are suggested in more than one study. Briefly, all proteins identified as significantly regulated between benign and malignant subgroups and the above given proteins constituted a first selection (Appendix A). Among these, a subset of proteins was further selected based on whether they could be confirmed as proteins able to separate benign versus malignant OC tumors in one or more of the selected literature studies [15,16,17,18,22]. In total, 284 proteins, for which protein-specific peptides could be identified when matched against the human background proteome, were selected. For these, 388 synthetic unique peptides were ordered from JPT (Berlin, Germany). Two peptides were ordered from 104 proteins and one peptide was ordered from 180 proteins, respectively, based on estimated peptide quality. Of the 388 peptides, 185 peptides from 177 proteins were selected for the final scheduled SRM assay with a minimum of four transitions per peptide based on their performance in the plasma background (transition list can be found in the PASSEL repository) [23] with ID PASS01544. No addition of peptides has been done to compensate for unidentified proteins, in line with the purpose of the study.

### 2.6. Selected Reaction Monitoring Mass Spectrometry

The SRM measurements were performed on a TSQ Quantiva triple quadrupole mass spectrometer (Thermo Fisher Scientific, San Jose, CA, USA) equipped with a nano electrospray ion source (EASY-spray, Thermo Fisher Scientific, San Jose, CA, USA). TSQ global parameters were: polarity: positive; spray voltage: 2000 V; and transfer tube temperature: 325 °C. TSQ scan parameters were: quadrupole resolution: 0.7 FWHM in both Q1 and Q3; cycle time: 1.7 s; CID gas (argon) 2 mTorr; source fragmentation: 10V; Chrom filter: 3 s; and 1084 transitions were measured simultaneously. Peptides were loaded on an Easy-nLC II system (Thermo Fisher Scientific, San Jose, CA, USA) with 15 cm, 3 μm columns (Thermo Fisher Scientific, San Jose, CA, USA). Peptides were eluted with a linear gradient from 95% solvent A (0.1% formic acid in water) and 5% solvent B (0.1% formic acid in acetonitrile) to 35% solvent B over 60 min at a constant pressure of 280 bars and a flow rate of 300 nL/min. Retention times of the scheduled LC-SRM assays were calibrated by measuring the iRT-peptides (Biognosys, Schlieren, Switzerland) in non-scheduled LC-SRM [24]. The resulting data were analyzed in the Anubis SRM analysis software v.1.4.4 with a q-value of 0.05 [25]. The SRM data were visualized for QC purposes using the software Skyline v3.6.0.10162 (MacCoss lab, University of Washington, Seattle, WA, USA). Only peptides with stable performance of both the synthetic and native peptide across samples were included. Data from synthetic peptides spiked into plasma background were used to create the reference file. Data were plotted in GraphPad (Prism) and linear regression was performed including all data points down to a concentration when linearity would be greatly obscured by including that concentration or providing a linearity below 0.5. No weighting was used. The raw data were deposited and are available at the server: https://db.systemsbiology.net/sbeams/cgi/PeptideAtlas/PASS_View (accessed on 10 November 2021) or access www.peptideatlas.org with dataset ID PASS01544 [23].

Subsequently, Loess Global normalization was performed using Normalyzer v.1.0 (Dep. of Immunotechnology, Lund University, Lund, Sweden) and all further analyses were done on normalized data [26]. The normalized data can be seen in Appendix A. After QC of depleted proteins, peptides for the 11 proteins were removed from the analysis set and all subsequent data analyses have been performed on 166 peptides from 159 proteins, which were identified in 85% or more of all samples. Nine samples had one missing value, three samples had five to eleven missing values and the remaining 62 samples had no missing values.

### 2.7. Multivariate Analyses

To evaluate differences between subgroups included in the study, both multivariate unsupervised and supervised approaches were used. Principal component analysis (PCA), t-Distributed Stochastic Neighbor Embedding (tSNE) and heatmaps with hierarchical clustering, (Qlucore Omics Explorer v.3.5, Qlucore, Lund, Sweden) [27], were used for initial data exploration to assess sample and variable separation and distribution in the investigated groups. PCA was assessed using floating levels of variance (for noise reduction) and *p*-values (two-group *t*-test, assuming no difference in variance) for visualization of the different subgroups (examples in Appendix A). *p*-values < 0.05 are reported in the Appendix A, but during the data exploration, no fixed *p*-value was used. The number of overlapping peptides between subgroup comparisons with cut-off of *p* < 0.05, is illustrated in Venn diagrams. The main tool for analysis, supervised orthogonal projection to latent structures—discriminant analysis (OPLS-DA), was used to compare subgroup separation using predictive (Q2-value) discriminant models based on all or subsets of peptides [28]. In the OPLS-DA, the parameters R2X(cum), R2Y(cum), and Q2(cum) describe the fit of each model. The predictive value Q2(cum) is based on seven-fold cross validation and the value varies between 0 to 1, where a high Q2 value (>0.5) is indicative of good prediction for the specific model. However, there is no sharp cutoff for the Q2 value. In this study, Q2 values were used to compare how well groups were separated. For each subgroup comparison, the peptides identified as variables of importance for the projection (VIP) of the model (defined as VIP values above 1) in the initial model calculation including all peptides, were selected and an OPLS-DA model was calculated based on peptides with VIP > 1. As a quality control, permutation plots (with 100 permutations per model, used to assess the risk that the model just fits the training set but cannot be used for prediction of new observations) was performed to assess the risk of a model being spurious. None of the models with Q2 > 0.3 were found to be spurious.

### 2.8. Evaluation of Predictive Value vs. p-Value

The peptides with *p* < 0.05 in the unsupervised analysis and with a predictive Q2 > 1 in the supervised analysis were assessed for the number of coinciding peptides within each subgroup comparison. To create a simple overview of the different analysis approaches, OPLS-DA and *p*-values in PCA, and compare these, a color (heatmap) ranking was done in Excel using the predictive Q2 values for the models with VIP > 1 and number of peptides with *p* < 0.05 (Appendix A).

## 3. Results

### 3.1. Main Results

The primary aim of this study was to investigate whether proteins identified in OC cyst fluids and primary tumor tissues, with the potential to separate benign ovarian tumors and OC histology subgroups, could also be identified in OC patient plasma samples and could be used for creating predictive models for differentiating tumor subtypes (Figure 1). Protein-specific peptides were analyzed in a cohort of benign and malignant ovarian tumors of different histology subtypes and stages (benign *n* = 16; borderline type stage I, *n* = 13; malignant stage I–II, *n* = 28; malignant stage III, *n* = 17) (Table 1).

#### 3.1.1. Histology Specific Peptide Patterns

In total, 177 selected proteins were analyzed by targeted SRM MS in plasma. Out of these, 16 peptides from 11 proteins were derived from abundant plasma proteins as a quality control of the plasma depletion. All samples were satisfactorily depleted. Of the 166 peptides included in the further analyses, 91 peptides had a *p*-value < 0.05 in any pairwise benign to OC subgroup comparisons using PCA. Comparing the group of less common OC, including borderline type tumors to the more abundant HGSC, it was clear that few peptides with *p*-values < 0.05 were shared between the two groups and the majority of peptides with *p* < 0.05 were specific to each comparison (Figure 2a). The three separate comparisons of early stage endometrioid, mucinous and LGSC also had few common peptides with *p* < 0.05 while most peptides were unique (Figure 2b). Similarly, for the serous OC of different stages and grade, the majority of peptides were unique, and few were shared (Figure 2c). In summary, PCA renders the different histological subtypes of OC included in this study highly heterogenous, as measured by a highly multiplexed SRM assay.

The targeted SRM MS analysis data of plasma samples from benign ovarian tumors vs. OC were analyzed using both supervised and unsupervised multivariate approaches. The unsupervised approaches were used for data exploration (Appendix A). These analyses revealed histology-specific peptide profiles in OC and in benign samples (Table 2). Simple overviews of the supervised OPLS-DA subgroup analysis models A–C are found in Figure 3a. Comparing the benign group to all malignant samples as one group with OPLS-DA resulted in a poor predictive model for separation, with and without the borderline type tumors included in the OC group Q2 = 0.138 (model A1) and Q2 = 0.226 (model A2). In this supervised analysis, between 55–77 of the 166 peptides in the SRM assay contributed to the separation of each model (column 5, Table 2). Despite 30 and 41, respectively, peptides having *p* < 0.05, the OPLS-DA analyses still provided poor separation, in the respective comparison (A1 and A2; Table 2, last column). All OPLS-DA analysis is also in Appendix A.

The OC were then analyzed in groups based on FIGO 2014 grade and stage and histology (Table 1). Comparing benign to LGSC, endometrioid and mucinous OC together provided a poor predictive value of Q2 = 0.221 when including borderline type tumors. However, a good predictive value of Q2 = 0.624 was obtained when excluding borderline type tumors (model B1 and B2; Figure 3b). The benign vs. borderline type tumors alone also achieved poor predictive separation, Q2 = 0.247 (model B2.4). Taken together, these analyses demonstrate that borderline type tumors are similar to benign adenomas as they cannot successfully be separated from these by the targeted SRM MS assay. Based on this initial analysis, the borderline type tumors were excluded from further sub-group analyses.

When designing the SRM MS assay in this study, proteins were selected from studies focusing on early-stage disease including all OC histologic subtypes. Early-stage OC (stage I-II) as one heterogenous group including all histological subtypes (endometrioid, mucinous, LGSC and HGSC) could not be adequately separated from the benign tumors, Q2 = 0.369 (model E1, Table 2). However, when separating the early-stage (I-II) OC into each histological subgroup and comparing these to benign separately, the predictive value increased for all differential diagnoses; mucinous Q2 = 0.611 (model B2.2), endometrioid Q2 = 0.428 (model B2.1/C1.1), LGSC Q2 = 0.615 (model B2.3) and HGSC Q2 = 0.513 (model C2.2; Figure 3c). The benign tumors, compared to HGSC in late stage (stage III), resulted in a predictive value of 0.430 (model C2.3; Figure 3d). With the addition of HGSC stage I-II, the Q2 value decreased to 0.375 (model C2).

#### 3.1.2. Multivariate Analysis of the Ovarian Cancer Subgroups

The histology-specific diversity in plasma was also evident when analyzing the different OC subgroups alone, excluding the benign tumors from the OPLS-DA analysis. Models D and E are found in Table 3. Comparing the less common low-grade OC (LGSC, mucinous, endometrioid stage I-II) versus high-grade OC, with and without borderline type tumors included in the low-grade OC group, provided weak separations, Q2 = 0.175 (model E4) and Q2 = 0.356 (model E5), respectively. Likewise, comparison of all early-stage OC with late-stage OC with or without borderline type tumors included to the early-stage OC provided fairly weak separations, Q2 = 0.374 (model E7) and Q2 = 0.299 (model E8). This created relatively poor predictive models which could be anticipated based on the inherent molecular differences within the low-grade OC group and high-grade OC. However, comparisons of different histologic subgroups (Table 3) provided higher predictive values for the LGSC vs. mucinous, Q2 = 0.468 (model D1) and LGSC vs. endometrioid, Q2 = 0.654 (model D3), confirming that the distinct molecular differences within the histologic subgroups increase group heterogeneity, which explains why modelling such diverse groups as one entity fails to provide clear separation.

All early-stage OC were compared to all late-stage OC, with and without borderline type tumors. The separation from stage III was increased when borderline type tumors were included with the stage I–II OC, which may be due to the similarities between the benign and the borderline type tumors, Q2 = 0.374 (model E7) and Q2 = 0.299 (model E8). OPLS-DA analysis of different stages of the same histological subtype provided good predictive models; early-stage HGSC vs. late-stage HGSC, Q2 = 0.592 (model D10; Figure 4), as well as early-stage LGSC and HGSC vs. late-stage HGSC, Q2 = 0.606 (model E6; Figure 4) indicative of a distinct heterogeneity between stage I–II and stage III disease in plasma. Further division into the different histologic subgroups increased the predictive model for LGSC alone vs. late-stage HGSC, Q2 = 0.634 (model D7; Figure 4).

In summary, we found a detectable heterogeneity between grade, stage I–II and stage III disease in plasma with the multiplexed SRM MS assay. The study identifies subgroup-specific peptides and demonstrates how subgroup-specific classification improves the separation of each OC histology in plasma samples. Better separation between groups was achieved when samples were stratified into grade, stage and histology compared to all malignant versus all benign tumors (summarized in Figure 3a, Table 2 and Table 3). The best separation was obtained for early-stage OC vs. benign tumors, which could be expected, as patients with early-stage disease were represented more in the original cyst fluid selection of proteins and intentionally used to build the basis for this study.

## 4. Discussion

Candidate diagnostic biomarkers for OC have mainly been primarily discovered from studies of serum or plasma dominated by the high-grade serous ovarian cancer subtype in late stages. This has provided a selection of HGSC late-stage-specific biomarkers [15,29,30]. The wide-spread use of cancer antigen 125 (MUC 16/CA125) is a good example of such a biomarker, having acceptable sensitivity for HGSC but only in late-stage OC and preferably in women over 50 years of age [6]. New diagnostic or screening biomarkers for early-stage disease are vital to lower mortality in OC. To overcome this problem, we scrutinized not only discovery sets with traditional OC composition cohorts but also biomarkers discovered in early-stage low-grade (LGSC, mucinous, endometrioid) and HGSC cyst fluids and tumor tissues [12,22]. The study population in this study included benign tumors and a large group of early-stage OC of low-grade, which we could compare to HGSC in both early and late stages. With this approach, we demonstrate that protein composition in plasma samples representing these histological subtypes and stages is diverse and peptide signatures are indeed different for each subtype. Multivariate analysis separate benign tumors from each OC histological subtype with better accuracy than the combined OC cohort. The same is shown for separation within each histologic subgroup and stages.

Ovarian cancer is classified into five major histotypes, including HGSC, endometrioid, mucinous, clear cell and LGSC, where HGSC is by far the most common. These histotypes differ with respect to appearance, genetic molecular alterations, therapy response and outcome [31,32]. While immunohistochemistry (IHC)-based classification of tumor biopsies is the most superior and efficient diagnostic technique today, the ability to stratify each histologic subgroup with the help of a simple blood test using a protein signature has several important advantages and potential uses. Pre-diagnostic knowledge of OC histology may affect the choice of surgical technique, neoadjuvant and adjuvant treatment. A blood test is convenient and would spare the patient from the procedure of acquiring a tumor biopsy, which in OC requires anesthesia and surgery. In addition, peptide/protein analysis can be both quick and cheap as compared to other emerging platforms such as the analysis of circulation tumor DNA with Next Generation Sequencing.

Numerous attempts to identify new diagnostic or screening protein biomarkers for OC can be found in the literature [33,34]. The goal is to increase survival of this deadly cancer by shifting the majority of newly diagnosed OC commonly found in stage III, when the disease has disseminated in the abdomen, to be identified in stage I, when the disease is still confined to its primary location (ovary or fallopian tube). Discovery cohorts from cell lines, ovarian cyst fluids, ascites, urine, tumor tissue and, of course, serum or plasma have been used without consideration of the heterogenous composition of OC, and therefore biomarker signatures identified in such studies often fail in validation. We also have to consider that a pre-surgical diagnostic sample has to be easily accessed (blood, urine or cervical smear) and not impose any risk to the patient (fine needle aspiration of cysts). Our hypothesis, that with a combination of candidate proteins from discovery studies that focus on all OC subtypes, we can also define specific protein signatures for low-grade early-stage OC, was confirmed in this study. Benign tumors were clearly separated from all low-grade OC including mucinous, endometrioid and LGSC as well as HGSC in both early and late stages. Interestingly, in the protein signatures defining the LGSC, endometrioid and mucinous early-stage subgroup zero proteins were shared between these groups.

Large discovery studies on the gene or protein level not only discover single mutations or proteins for diagnostic or therapeutic purposes, but common histologic traits can be used to find gene or protein signatures that increase the possibility to characterize and personalize each patient’s disease and treatment. For HGSC, molecular subtype classification through novel signatures and pathways has been suggested based on studies within The Cancer Genome Atlas (TCGA) database [35]. This was followed by the Clinical Proteomics Tumor Analysis Consortium (CPTAC) analyses of 169 HGSC tumors by protein expression and correlated the results with the TCGA database to identify novel signatures and pathways [36]. A study of the proteomes of OC and fallopian tube cell lines could define three separate groups, epithelial, clear-cell and mesenchymal. They also present a HGSC prognostic 67-protein signature for good or poor overall survival [37,38]. MS-MS was used to identify a new biomarker to separate endometrioid from HGSC by immunohistochemistry, but authors suggest that a multiplex panel should be used [39].

Targeted SRM analysis provides several advantages and features important for clinical application. An SRM assay can be made highly specific through ensuring the selection of correct peptides (protein specific) and analyzing an adequate number of transitions. It can be seamlessly multiplexed with hundreds of variables (peptides). In addition to high specificity, SRM analysis provides sensitivity and linearity that offers the opportunity to accurately quantify peptides. Absolute quantification is also possible to achieve through the use of isotope-labeled synthetic peptides. Compared to many other types of mass spectrometers, the instrumentation used for this type of analysis is relatively simple and robust. Benchtop mass spectrometers have been used in clinics for decades to analyze small molecules. Therefore, it is realistic to believe that a peptide SRM assay, as the one developed here, could find its way to a clinical setting. Additionally, with the use of “Next Generation Proteomics” like targeted SRM MS, an increasingly large number of different proteins can be detected in only 5–20 µL plasma. This can be compared to the need for 10–20 mL plasma for the detection of mutations in circulating cell free tumor DNA (ctDNA) in early-stage cancer. Proteomic technologies and protocols have the potential to be directly integrated into current clinical work flows since they are rapid, easy-to-use, robust and economical [40].

In this quite complex study setup with numerous OC subclasses, multivariate discriminant analysis (MVA) was used, as our aim was not to reduce the number of included peptides in the analysis to an absolute minimum, but to use the possibilities of simultaneous collection of hundreds of peptide values using SRM MS. One of the strengths with MVA is that the result depends on the interaction and performance of numerous interdependent variables, instead of relying on the readout of one or few variables, where their interaction is of no importance (as with *p*-values). An important step in a validation study would be to use a selection or ranking system, to decide which subgroup a random sample would belong to, depending on the readout of approximately 100 to 150 peptides of interest.

Limitations of our study include a relatively small sample size in each histologic subgroup and the lack of patients with clear-cell OC and mucinous adenoma. However, this study should be considered more as an evaluation of the possibility to create histology-specific protein signatures and use these for subgroup classification. Discriminatory single biomarkers for this purpose are not likely to be found. With the area of proteomics moving quickly towards high throughput, high-precision multiple target assays in small volumes of plasma samples will be able to pursue these findings. The second limitation, due to the study design with focus on low-grade OC in stage I–II and borderline type tumors, is that the study population does not reflect a normal setting of OC presented in the clinic, which would be required in a proper validation study. Third, a prior selection of proteins may exclude ones that could have made an impact on the separation of groups. With this strategy, we could still validate our hypothesis, that benign tumors and OC histology subgroups can be separated by peptide signatures with unique peptide signatures specific for each histologic subgroup in primarily early-stage disease.

## 5. Conclusions

In this study we have used prior mass spectrometry discovery sets of OC biomarkers, found in the literature and in-house results from ovarian cyst fluid and tumor tissue proteomic studies, to explore plasma samples from different epithelial-derived ovarian histologic subgroups in primarily early-stage disease. We found that unique specified peptide signatures for each histologic subgroup had the highest predictive values, suggesting fast multiple targeted protein analysis will be part of the next generation diagnostic platform.

## Figures and Tables

**Figure 1 cancers-13-05713-f001:**
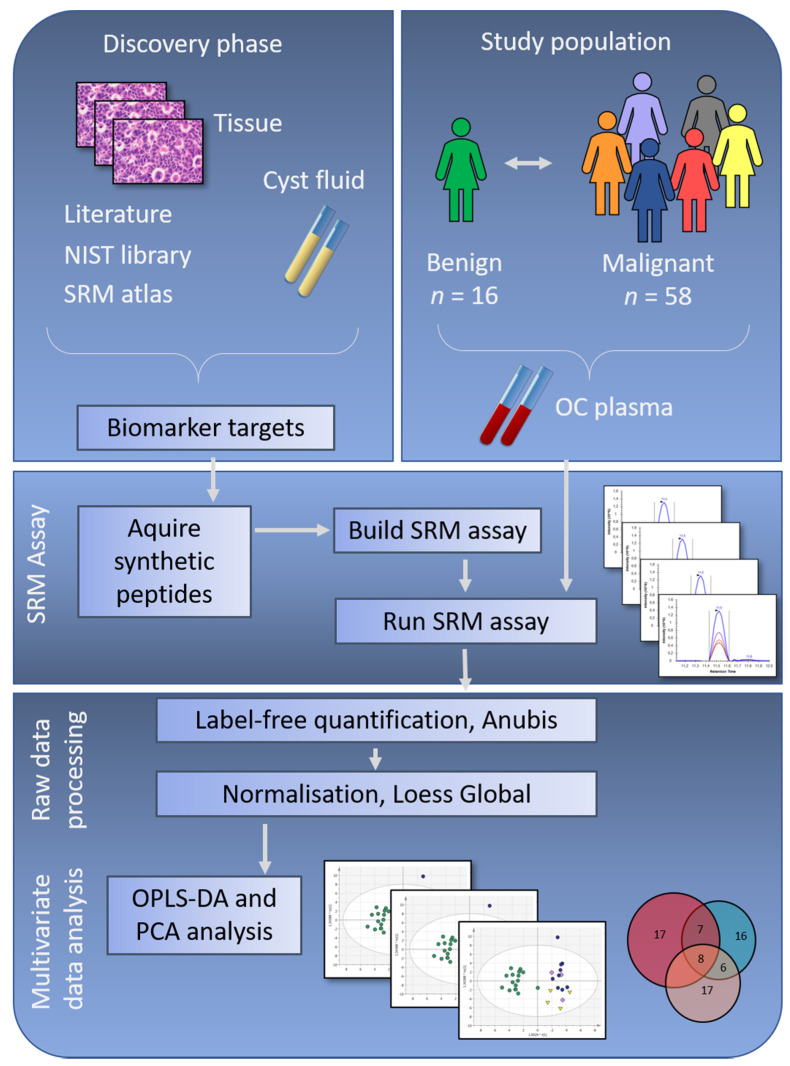
Study workflow showing the discovery phase with protein selection based on ovarian cancer (OC) tissue and cyst fluid, together with selected literature references and relevant data deposited in repositories. Meanwhile, the study population was selected, with inclusion of samples from benign, early-stage and late-stage OC. To verify if the potential biomarker targets identified in the discovery phase were possible to identify in patient plasma samples, synthetic peptides corresponding to the biomarker targets were ordered, and a selected reaction monitoring (SRM) assay was built. Peptides with stable identification in a group of pooled plasma samples were included in the final SRM assay, which was run on plasma samples from the complete study population. In the final step of this study, multivariate analyses were used to retrieve information on OC subgroup differences. NIST = National Institute of Standards & Technology, *n* = number of samples, OPLS-DA = Orthogonal Projection to Latent Structures - Discriminant Analysis, PCA = Principal Component Analysis.

**Figure 2 cancers-13-05713-f002:**
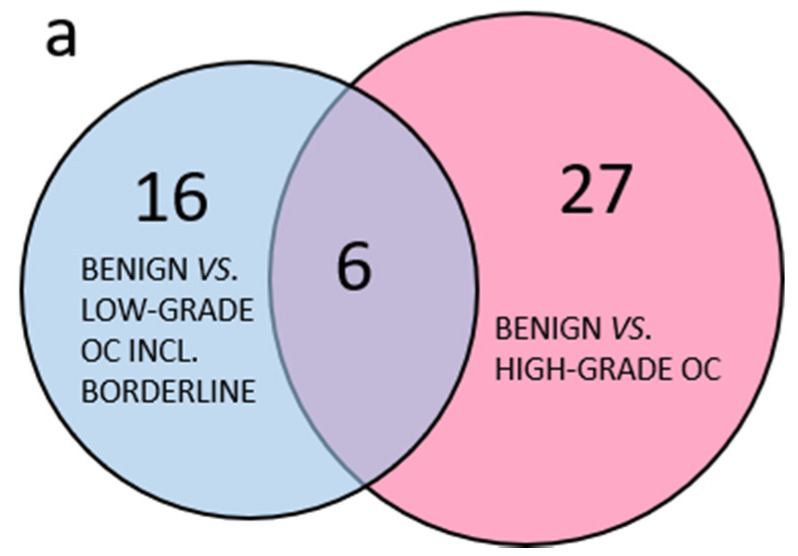
Venn diagram of benign and OC subgroups with number of overlapping peptides with *p* < 0.05 (*t*-test). (**a**) Benign vs. less common low-grade OC including borderline type tumors compared to benign vs. high-grade OC; (**b**) comparison of benign vs. OC low-grade subgroups; (**c**) comparison of benign vs. all included low-grade OC, HGSC stage I–II or HGSC stage III. 3.1.2. Multivariate Analysis of Benign Versus Malignant Tumors.

**Figure 3 cancers-13-05713-f003:**
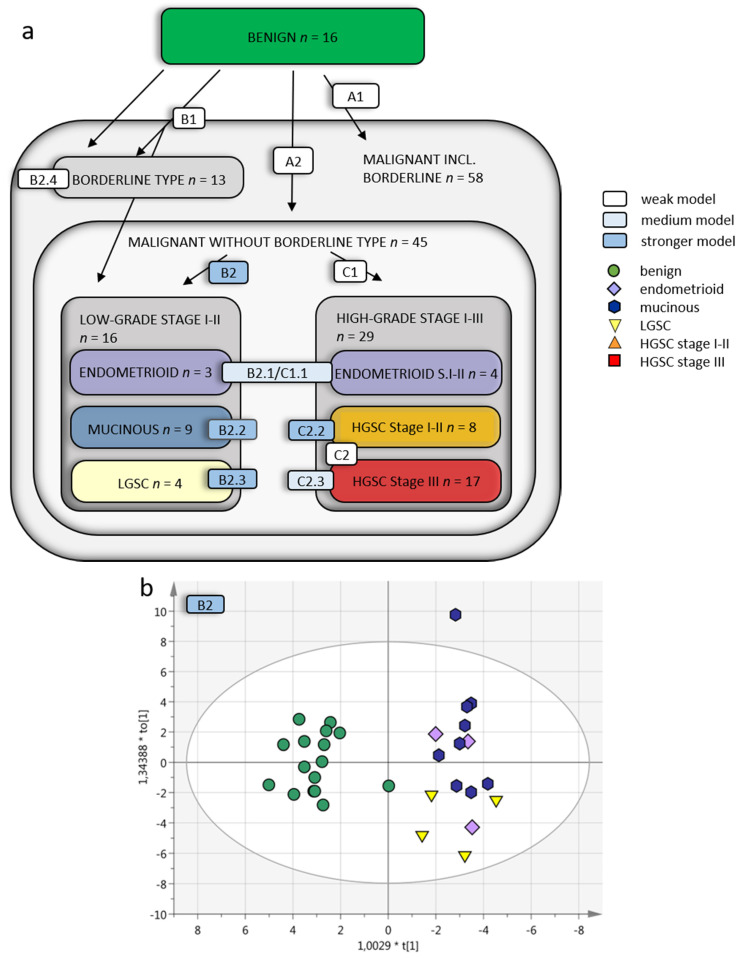
Multivariate OPLS-DA models of OC subgroups. (**a**) Overview of included OC subgroups. The models with higher predictive Q2 values are seen when the OC malignant samples are further divided into subgroups depending on grade, histology and stage. Q2 is the predictive value of the model and varies between 0–1, where > 0.5 indicates a good predictive value for the model. (**b**) OPLS-DA model B2 of benign vs. low-grade OC stage I–II samples (without borderline type tumors) with VIP > 1 (59 variables) and a Q2 value of 0.624, indicating reliable predictability. The individual low-grade samples have been colored according to subgroup for visualization purposes only but were grouped in the analysis. (**c**) Benign vs. HGSC stage I–II (OPLS-DA model C2.2) had a Q2 of 0.513 and (**d**) benign vs. HGSC stage III (model C2.3) had a Q2 of 0.43. For visualization, additional components have been added in c–d, as the optimal Q2 value had 1 + 0 + 0 components. OPLS-DA = orthogonal projection to latent structures-discriminant analysis, OC = ovarian cancer, VIP = variables of importance for projection of plot, LGSC = low-grade serous cancer, HGSC = high-grade serous cancer.

**Figure 4 cancers-13-05713-f004:**
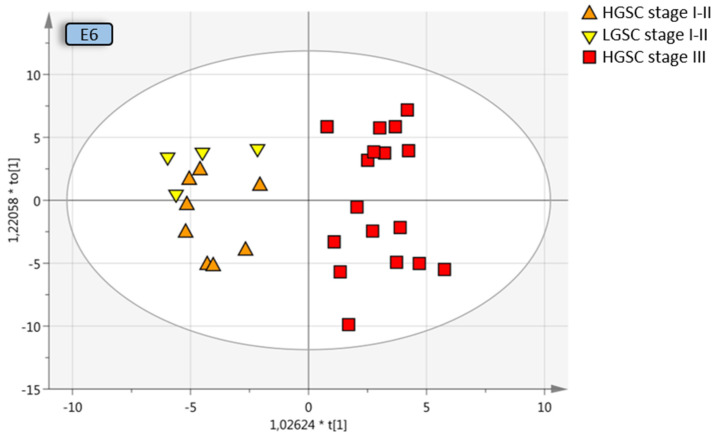
Multivariate OPLS-DA model where LGSC and HGSC stage I–II are grouped and compared to HGSC stage III, resulting in a predictive value of 0.612 (model E6, Table 3). Coloring has been done according to subgroup and not the OPLS-DA classes for visualization purposes. OPLS-DA = orthogonal projection to latent structures-discriminant analysis, LGSC = low-grade serous cancer, HGSC = high-grade serous cancer.

**Table 1 cancers-13-05713-t001:** Study population overview.

Histology	*n*	Grade ^1^	Stage ^1^	Mean Age (Years)	SD Age (Years)
Serous benign	16	N/A	N/A	69	8.7
Serous borderline	13	N/A	I	51.7	18.4
LGSC	4	low	I–II	52.3	5.7
Mucinous	9	low	I–II	67.3	7.3
Endometrioid ^2^	3	low	I–II	76.3	9.1
Endometrioid ^2^	4	high	I–II	59	16.2
HGSC	8	high	I–II	59.3	5.1
HGSC	17	high	III	61.5	12.1

^1^ According to FIGO (International Federation of Gynecology and Obstetrics 2014). ^2^ All endometrioid OC, *n* = 7, mean age is 66.4 (±15.7) years. *n* = number of samples, N/A = not applicable, LGSC = low-grade serous cancer, HGSC = high-grade serous cancer.

**Table 2 cancers-13-05713-t002:** Overview of comparisons of benign vs. malignant subgroups including borderline type tumors.

Benign vs. Subgroup	*n*	OPLS-DA Model-Name (also Figure 2a)	Q2, VIP > 1.0	#Peptides VIP > 1.0 Unique ^1^ Peptides	#Peptides *p* < 0.05 (% Shared with VIP > 1)Unique ^2^ Peptides
All malignant incl. Borderline	74	A1	0.138	63	30, q = 0.26
All malignant excl. Borderline	61	A2	0.226	61	41, q = 0.184
Low-grade ^1^ incl. Borderline	45	B1	0.221	63	22, q = 0.298
Low-grade ^1^ excl. Borderline	33	B2	0.624	59	37, q = 0.219
Borderline type tumor	29	B2.4	0.247	65, ^1^ 0	11 (45%) ^2^ 6 q = 0.688
Stage I–II (all histologies)	44	E1 (B2 + C1.1 + C2.2)	0.369	66	39, q = 0.194
Endometrioid stage I–II	20	B2.1/C1.1	0.428	61, ^1^ 5	17 (100%) ^2^ 5 q = 0.447
Mucinous stage I–II	25	B2.2	0.611	77, ^1^ 8	41 (95%) ^2^ 9 q = 0.190
LGSC stage I–II	20	B2.3	0.615	63, ^1^ 15	11 (100%) ^2^ 5 q = 0.567
High-grade ^1^	45	C1	0.333	55	33, q = 0.234
HGSC stage I–III	41	C2	0.375	59	42, q = 0.193
HGSC stage I–II	24	C2.2	0.513	62, ^1^ 9	34 (94%) ^2^ 9 q = 0.206
HGSC stage III	33	C2.3	0.43	65, ^1^ 8	39 (95%) ^2^ 12 q = 0.212

For the subgroups listed in Table 1 (endometrioid as one group), comparisons have been made to the benign group as follows; total number of peptides included in comparison (peptides with VIP > 1), ^1^ number of unique peptides in that comparison, ^2^ number of unique peptides with *p*-value < 0.05 that also are part of the VIP > 1 peptides (in %) for the specific subgroups. For more information see Appendix A. OPLS-DA = orthogonal projection to latent structures-discriminant analysis, Q2 = predictive value, VIP = variables of importance for the projection, ^1^ as defined in Table 1.

**Table 3 cancers-13-05713-t003:** Comparison between malignant subgroups.

Subgroups in Comparison	*n*	OPLS-DA Model	Q2, VIP > 1.0	#Peptides VIP > 1.0	#Peptides *p* < 0.05
LGSC stage I–II vs. Mucinous stage I–II	13	D1	0.468	71	25, q = 0.313
Endometrioid stage I–II vs. Mucinous stage I–II	16	D2	0.356	69	14, q = 0.465
Endometrioid stage I–II vs. LGSC stage I–II	11	D3	0.654	66	7, q = 0.761
LGSC stage I–II vs. HGSC stage I–II	12	D4	0.41	72	9, q = 0.569
Endometrioid ^1^ stage I–II vs. HGSC stage I–II	15	D5	0.354	70	9, q = 0.579
Mucinous stage I–II vs. HGSC stage I–II	17	D6	0.356	63	17, q = 0.436
LGSC stage I–II vs. HGSC stage III	21	D7	0.634	65	25, q = 0.302
Endometrioid ^1^ stage I–II vs. HGSC stage III	24	D8	0.388	61	25, q = 0.33
Mucinous stage I–II vs. HGSC stage III	26	D9	0.379	60	10, q = 0.658
HGSC stage I–II vs. HGSC stage III	25	D10	0.592	66	18, q = 0.404
Low-grade incl. Borderline vs. High-grade	58	E4	0.175	57	24, q = 0.329
Low-grade vs. High-grade	45	E5	0.356	63	16, q = 0.489
LGSC and HGSC stage I–II vs. HGSC stage III	29	E6	0.612	69	34, q = 0.241
All stage I–II incl. borderline vs. stage III	58	E7	0.374	60	47, q = 0.168
All stage I–II vs. stage III	45	E8	0.299	57	29, q = 0.272

^1^ Endometrioid subgroup, *n* = 7. Low-grade = LGSC, mucinous and endometrioid OC. High-grade = HGSC and HG endometrioid OC.

## Data Availability

The data presented in this study are openly available and the raw data and transition list have been deposited in PASSEL and are available at the server: https://db.systemsbiology.net/sbeams/cgi/PeptideAtlas/PASS_View (accessed on 10 November 2021) or www.peptideatlas.org, with dataset ID PASS01544. [23].

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
