# Peer review of "Targeted Selected Reaction Monitoring Verifies Histology Specific Peptide Signatures in Epithelial Ovarian Cancer"

_cancers, 2021, doi:10.3390/cancers13225713_

Round 1

Reviewer 1 Report

This paper have used a proteomics based approach to identify peptides enriched for each ovarian cancer histological subtypes from plasma isolated from each patient subgroups. Furthermore, the authors have illustrated histology specific peptide patterns through multivariate PCA and OPLS-DA analysis and highlighted that they could derive predictive scores when comparing peptides within and between subgroups and potentially use them for characterizing ovarian cancer subtypes. However, I feel that certain key experiments are still missing to strengthen the observed molecular signaling, which should be addressed by the authors.

  1. The results of multivariate analysis in Figure 3 & 4 shows clear distinction of peptide clusters between subgroups and their predictive value. However, other than listing of number of peptides, significance or biological relevance of these peptides are not discussed at all. Here, the authors need to further explore on the nature of peptides enriched in each comparisons, identify functional protein networks and their biological relevance. For example, does peptides of late stage HGSC biomarkers such as CA125 and HE4 show up in their analysis? If no, what other missed peptides of validated proteins stand out in their analysis?
  2. In the beginning of the manuscript the authors have identified that lack of biomarkers for early stage disease lack specificity and sensitivity. In contrast, in section 3.1.2, they have excluded borderline type tumors from further analysis. This contradicts one of their objectives proposed in the introduction. Perhaps, if they could perform SRM-MS analysis from healthy donors with no benign tumors could help them identify peptide signature that is exclusive for benign tumors and then perform subsequent sub-group analysis.

Author Response

RESPONSE TO REVIEWER 1

We thank the reviewer for the comments to our work. A native English-speaking colleague have read the manuscript and corrections have been performed according to her suggestions in the revised manuscript. These are our responses to the comments provided by reviewer 1:

  1. We agree with the reviewer that in an exploratory discovery analysis it is of utmost importance and interest to undertake functional network analysis and in-dept evaluate the nature and biological relevance of proteins analyzed. However, current work is differently designed and is instead strictly analytical, evaluating a highly selected peptide subset in depleted blood. Therefore, a pathway analysis cannot provide any biologically meaningful information. The depletion was performed to allow for analysis of a subset of the free peptidome remaining after removing abundant serum proteins, which would otherwise be masked by higher abundance proteins. It should also be pointed out that it is well known that albumin and other abundant serum proteins are “sticky” and bind many proteins. A depletion of these will therefore also deplete for proteins bound to these serum proteins. However, what we see in our analyses is that the depletion itself is highly reproducible why the remaining peptides that we analyze should still be reproducible for the proof of concept that we are claiming with this study. With regard to CA125 and HE4, these glycoproteins were discovered in HGSC dominated cohorts and have thus best diagnostic impact on HGSC, while both of these biomarkers is not produced by i.e. the mucinous histotype. In this manuscript we wanted to look beyond the classical biomarkers to address the diagnostic potential of different histologic subtypes of OC. To validate the sensitivity and specificity of a single biomarker the study-cohort also need to be much larger.
  2. In the analysis where the borderline group was included, the benign and borderline groups were demonstrated to be very similar, based on the peptide selection. In analysis where the borderline samples were analyzed together with the ovarian cancer subgroups, poorer predictability was achieved, which also points to the conclusion of this study; ovarian cancer subgroups are heterogenous and should not be grouped for optimal identification of predictive peptides. The resemblance in the plasma proteome between benign ovarian cysts and borderline type tumor cyst has been noticed before by us and others. For differential diagnostic purposes, borderline type tumors are fairly simple to separate from benign simple cysts with specialized transvaginal ultrasound (TVU) available at gynecology cancer units.

A next and extended study would need larger sample sets and could as the reviewer suggests also include samples for healthy donors to better define the benign subtype. It is, also important to remember that the biomarkers searched for in this study is NOT for population screening but rather for the differential diagnosis of cysts and tumors discovered in ordinary imaging analysis such as TVU or CT-scans.

Reviewer 2 Report

A spectral library was built into the SRM management software Skyline

1) The author should describe more in detail how the library was built and which parameters were used.

 The list of target proteins generated from our previous study [10-14] and literature 128 [15-18,20] was imported to and in-silico digested in Skyline

2) The criteria used for creating the protein panel need to be described in more detail.

Two peptides were ordered from 104 proteins 142 and one peptide was ordered from 180 proteins respectively, based on estimated peptide quality

3)How was the peptide quality estimated?

4)The authors should provide the normalized data that were used as an input for the PCA and OPLS-DA analyses.

“Of the 166 236 included peptides in the further analyses, 91 peptides had a p-value <0.05 in any pairwise benign to OC subgroup comparisons using PCA.....”

5)PCA is typically used for data exploration and dimensionality reduction but it doesn’t test a null hypothesis and generate a p-value. These p-values are the result of a t-test between each pair of sample groups, how is PCA used in this analysis?

6)In Figure 2 the names for the OPLS-DA models are used but the OPLS-DA analysis is not used in this figure and the model names are confusing.

The targeted SRM-MS analysis data of plasma samples from benign ovarian tumors vs. OC were analyzed using both supervised and unsupervised multivariate approaches.

7)I only see a supervised approach used (OPLS-DA) which unsupervised approach was used here?

8)When multiple peptides per proteins were measured what was the concordance of the peptides from the same protein being identified as discriminant in the various OPLS-DA analyses?  

This was followed by the Clinical Proteomics Tumor Analysis Consortium (CPTAC) analyses of 169 HGSC tumors by protein expression, and correlated the results with the TCGA database to identify novel signatures and  pathways [34]. A study of the proteomes of OC and fallopian tube cell lines could define  three separate groups, epithelial, clear-cell and mesenchymal. They also present a HGSC prognostic 67-protein signature for good or poor overall survival [35,36]. MS-MS was used  to identify a new biomarker to separate endometrioid from HGSC by immunohistochem-istry, but authors suggest that a multiple panel should be used [37]

9)In the discussion the authors talk about several proteomic studies that profiled OC samples. Were any of the proteins highlighted in these studies as potential marker, between tumor groups, included in this study and did the OPLS-DA analysis recapitulate any of the published findings (identified the relative peptides as significant between the relative groups)?

Author Response

RESPONSE TO REVIEWER 2  (for additional figures see the uploaded version)

We thank the reviewer for the comments to our work. A native English speaking colleague have read the manuscript and corrections have been performed according to her suggestions in the revised manuscript. These are our responses to the comments provided by reviewer 2:

A spectral library was built into the SRM management software Skyline

  • The author should describe more in detail how the library was built and which parameters were used.

Page 3, Line 129. The procedure for creating and importing a spectral library is now described through added references.

(MacLean, Tomazela et al. 2010, Teleman, Hauri et al. 2017)

MacLean, B., et al. (2010). "Skyline: an open source document editor for creating and analyzing targeted proteomics experiments." Bioinformatics 26(7): 966-968.

       SUMMARY: Skyline is a Windows client application for targeted proteomics method creation and quantitative data analysis. It is open source and freely available for academic and commercial use. The Skyline user interface simplifies the development of mass spectrometer methods and the analysis of data from targeted proteomics experiments performed using selected reaction monitoring (SRM). Skyline supports using and creating MS/MS spectral libraries from a wide variety of sources to choose SRM filters and verify results based on previously observed ion trap data. Skyline exports transition lists to and imports the native output files from Agilent, Applied Biosystems, Thermo Fisher Scientific and Waters triple quadrupole instruments, seamlessly connecting mass spectrometer output back to the experimental design document. The fast and compact Skyline file format is easily shared, even for experiments requiring many sample injections. A rich array of graphs displays results and provides powerful tools for inspecting data integrity as data are acquired, helping instrument operators to identify problems early. The Skyline dynamic report designer exports tabular data from the Skyline document model for in-depth analysis with common statistical tools. AVAILABILITY: Single-click, self-updating web installation is available at http://proteome.gs.washington.edu/software/skyline. This web site also provides access to instructional videos, a support board, an issues list and a link to the source code project.

Teleman, J., et al. (2017). "Improvements in Mass Spectrometry Assay Library Generation for Targeted Proteomics." J Proteome Res 16(7): 2384-2392.

       In data-independent acquisition mass spectrometry (DIA-MS), targeted extraction of peptide signals in silico using mass spectrometry assay libraries is a successful method for the identification and quantification of proteins. However, it remains unclear if high quality assay libraries with more accurate peptide ion coordinates can improve peptide target identification rates in DIA analysis. In this study, we systematically improved and evaluated the common algorithmic steps for assay library generation and demonstrate that increased assay quality results in substantially higher identification rates of peptide targets from mouse organ protein lysates measured by DIA-MS. The introduced changes are (1) a new spectrum interpretation algorithm, (2) reapplication of segmented retention time normalization, (3) a ppm fragment mass error matching threshold, (4) usage of internal peptide fragments, and (5) a multilevel false discovery rate calculation. Taken together, these changes yielded 14-36% more identified peptide targets at 1% assay false discovery rate and are implemented in three new open source tools, Fraggle, Tramler, and Franklin, available at https://github.com/fickludd/eviltools . The improved algorithms provide ways to better utilize discovery MS data, translating to substantially increased DIA performance and ultimately better foundations for drawing biological conclusions in DIA-based experiments.

The list of target proteins generated from our previous study [10-14] and literature 128 [15-18,20] was imported to and in-silico digested in Skyline

  • The criteria used for creating the protein panel need to be described in more detail.

Clarified on page 3, starting at line 141. The selection of target proteins is now described more in details under section 2.5. We have added a table which provides the details of where, how and how many times a protein was detected and the final selection so it can be followed protein by protein. It can be seen from this document that each protein that has been selected has been thoroughly evaluated in many different sources, both our in-house data and in publication. Supplementary table 1.

Two peptides were ordered from 104 proteins 142 and one peptide was ordered from 180 proteins respectively, based on estimated peptide quality

3)How was the peptide quality estimated? 

Peptide quality was based on total intensity in MS1 spectrum and % of total spectrum intensity contributed by identified transitions in the MS2 spectrum. The top 3 unique peptides per protein identified with highest MS1 spectrum intensity were selected. Out of these, the 1-2 peptides with the cleanest MS2 spectrum, meaning the highest % of total MS2 spectrum intensity contributed by identified peptide fragments were selected. For any protein where two peptides meeting the criterium of five good transitions (y- and b- ion signals) were selected for ordering a synthetic peptide. For any protein where only one peptide met these criterium, only one peptide was selected. Peptide identity and quality and final transition selection was subsequently verified using the synthetic peptides.

We can include above clarification in the manuscript if the reviewer thinks this is required. We have deliberately kept the SRM assay development short as this is done according to standard procedure and nothing in the details of this part is in scope of the aim or outcome of this work.

4)The authors should provide the normalized data that were used as an input for the PCA and OPLS-DA analyses.

The normalized data has been provided in Supplementary table 2.

“Of the 166 236 included peptides in the further analyses, 91 peptides had a p-value <0.05 in any pairwise benign to OC subgroup comparisons using PCA.....”

5) PCA is typically used for data exploration and dimensionality reduction but it doesn’t test a null hypothesis and generate a p-value. These p-values are the result of a t-test between each pair of sample groups, how is PCA used in this analysis?

The PCA was indeed used for data exploration and a “sanity check” of the data – both samples and peptide variable distribution with regards to outliers etc. -  prior to the supervised analysis. Both variance filtering (to reduce “noise”) and/or sliding p-values were used during this exploratory analysis to filter the number of peptides and expose the most important ones driving the analysis. Other plot types were also used for these initial views of the dataset, but the information of them is captured in the supervised data and the listings of variables with p-value <0.05. However, during the PCA, no sharp cutoffs were used, but for reporting purposes for this manuscript, the significance level of 0.05 was used. Well over 500 different plots has been viewed during the initial data exploration using PCA, tSNE, hierarchical clustering etc., so leaving the method out of the manuscript would not be correct. Nevertheless, adding plots of these numerous views would not be feasible, and the supervised plots are more suitable for visualizing purposes.

The text has been changed in section 2.7 (page 4, starting at line 193) and 3.1.2 (page 8, line 276) to include the above information and some illustrative examples from the data exploration process are added in Supplementary Figure SF1.

Supplementary Figure S1. Examples of plots from the initial data exploratory phase using Qlucore Omics tools (a) Principal component analysis of benign (blue) and mucinous (orange) samples, with a p-value filter for the two groups of p=0.01, q=0.07, resulting in 19 peptides (b) t-distributed Stochastic Neighbour Embedding (t-SNE) with benign tumors (blue) and low-grade Stage I ovarian cancer (green) with p=0.01, q=0.085, resulting in 19 peptides, (c) Hierarchical clustering of peptides, benign (blue) vs. malignant ovarian tumors, p=0.05 resulting in 41 peptides. Malignant cancers sorted according to group with low-grade (grey), high-grade stage I-II (orange) and HGSC stage III (bordeaux) (d) Hierarchical clustering of both samples and peptides of HGSC stage I-II vs. III resulting in 18 peptides at p=0.050 (e) Hierarchical clustering of both samples and peptides, benign tumors (blue) and HGSC stage III (bordeaux) with p<0.01, q=0.05, 14 peptides.

6)In Figure 2 the names for the OPLS-DA models are used but the OPLS-DA analysis is not used in this figure and the model names are confusing.

The model names were included as we thought that would give a more transparent overview of when the same groups had been compared. These names can easily be removed if required?

The targeted SRM-MS analysis data of plasma samples from benign ovarian tumors vs. OC were analyzed using both supervised and unsupervised multivariate approaches.

7)I only see a supervised approach used (OPLS-DA) which unsupervised approach was used here?

Thank you for pointing this out, examples had been overlooked. See also answer to question 5) above. The text has been changed in section 2.7 and 3.1.2 to include information and some illustrative examples from the data exploration process have been added in Supplementary Figure SF1. As written in the answer to Reviewer sub-question 5, only examples are shown as the supervised analysis captures the key features of the analyses and adding examples that represent the early data exploration process would not be doable (hundreds of plots).

8)When multiple peptides per proteins were measured what was the concordance of the peptides from the same protein being identified as discriminant in the various OPLS-DA analyses?  

There are both examples where both peptides from one protein is coming out in one specific OPLS-DA analysis and there are examples where they do not since there is always a cutoff in each analysis performed. In theory two peptides from one protein don’t necessarily need to behave exactly the same, both because of biological reasons (posttranslational modifications etc) and analytical reasons (behavior in MS analysis). This is why we consistently are referring to peptides and not proteins in this manuscript, as we only can refer to how a specific peptide behaves during the given circumstances. Please see below, correlation between two peptides for same protein below as an example (some with good others with poor correlation, at least without further transformation of the values).

This was followed by the Clinical Proteomics Tumor Analysis Consortium (CPTAC) analyses of 169 HGSC tumors by protein expression, and correlated the results with the TCGA database to identify novel signatures and  pathways [34]. A study of the proteomes of OC and fallopian tube cell lines could define  three separate groups, epithelial, clear-cell and mesenchymal. They also present a HGSC prognostic 67-protein signature for good or poor overall survival [35,36]. MS-MS was used  to identify a new biomarker to separate endometrioid from HGSC by immunohistochem-istry, but authors suggest that a multiple panel should be used [37]

9)In the discussion the authors talk about several proteomic studies that profiled OC samples. Were any of the proteins highlighted in these studies as potential marker, between tumor groups, included in this study and did the OPLS-DA analysis recapitulate any of the published findings (identified the relative peptides as significant between the relative groups)?

The uniqueness of the current study is the multivariate approach with a highly multiplexed SRM assay which allows for an analysis of contribution of many signals taken together. Also unique to this study is the histologic subtyping of the ovarian tumor samples where the most important finding and conclusion of this study is that these subgroups are truly unique and it is therefore important to further study larger sets of these subgroups and with multivariate approaches to allow for identifying clinically relevant signals. In this context, a single protein-to-protein comparison is not supporting neither the aim of or the conclusions of this work.

Reviewer 3 Report

Liljedahl and co-authors, starting from proteins identified by mass spectrometry -based proteomics in cyst fluids and ovarian tumor tissue, were able to construct a multiplex assay for simultaneous detection and relative quantification of 185 peptides from 177 proteins in plasma. The global aim of their work is to provide tolls for early detection of ovarian cancer that is still an unmet clinical need for this disease.

In their study they have included benign tumors as well as early stages tumors of different histology and grade. The main finding was that the heterogeneous group of early stages tumors could not be distinguished from benign tumors. However, this separation improved when the early stages tumors were distinguished in the histological subtypes and then compared to benign samples.

Although the approach is very interesting I think that the data are still too descriptive without any speculation about the proteins (among those selected from the discovery phase) and pathways that can be characteristics of the analyzed subtypes.

Furthermore discussion should be more focused also on previous data obtained by the authors’ them-self considering that in previous studies they found that cyst fluids are better than sera in early stage ovarian cancer diagnostics.

Author Response

RESPONSE TO REVIEWER 3

We thank the reviewer for the comments to our work. A native English-speaking colleague have read the manuscript and corrections have been performed according to her suggestions in the revised manuscript. These are our responses to the comments provided by reviewer 3:

We agree with the reviewer that functional network analyses and pathway analysis are powerful tools to understand the biological and thus clinical context of proteins in a multivariate analysis. This is something that is relevant in an exploratory discovery analysis for validating such work. However, current work is differently designed and is instead strictly analytical, evaluating a highly selected peptide subset in depleted blood. Therefore, a pathway analysis cannot provide any biologically meaningful information. The depletion was performed to allow for analysis of a subset of the free peptidome remaining after removing abundant serum proteins, which would otherwise be masked by higher abundance proteins. It should also be pointed out that it is well known that albumin and other abundant serum proteins are “sticky” and bind many proteins. A depletion of these will therefore also deplete for proteins bound to these serum proteins. However, what we see in our analyses is that the depletion itself is highly reproducible why the remaining peptides that we analyze should still be reproducible for the proof of concept that we are claiming with this study

The uniqueness of the current study is rather the multivariate approach with a highly multiplexed SRM assay which allows for an analysis of contribution of many signals taken together. Also unique to this study is the histological subtyping of the tumor samples where the most important finding and conclusion of this study is that these subgroups are truly unique, and it is therefore important to further study larger sets of these subgroups and with multivariate approached to allow for identifying clinically relevant signals. In this context, a single protein-to-protein comparison is not supporting either the aim of, or the conclusions of this work.

Liquid biopsies from ovarian cysts could definitely improve diagnosis of a cancer suspicious ovarian cyst, but as you probably are aware of, fine needle aspiration of a cancer cyst may increase the risk of cancer cell leakage to the abdomen, thus upstage the disease and affect survival for the individual patient. It is not recommended in the diagnostic work-up. That is why we focused the current study on analysis in a simple blood sample. A sentence to further clarify this topic has been added to the discussion Page 14, starting on line 415

Round 2

Reviewer 1 Report

The response provided by the authors suffice the issues I have raised earlier. I recommend the manuscript for publication

Author Response

Thank you.

Reviewer 2 Report

3)How was the peptide quality estimated? 

Peptide quality was based on total intensity in MS1 spectrum and % of total spectrum intensity contributed by identified transitions in the MS2 spectrum. The top 3 unique peptides per protein identified with highest MS1 spectrum intensity were selected. Out of these, the 1-2 peptides with the cleanest MS2 spectrum, meaning the highest % of total MS2 spectrum intensity contributed by identified peptide fragments were selected. For any protein where two peptides meeting the criterium of five good transitions (y- and b- ion signals) were selected for ordering a synthetic peptide. For any protein where only one peptide met these criterium, only one peptide was selected. Peptide identity and quality and final transition selection was subsequently verified using the synthetic peptides.

We can include above clarification in the manuscript if the reviewer thinks this is required. We have deliberately kept the SRM assay development short as this is done according to standard procedure and nothing in the details of this part is in scope of the aim or outcome of this work.

I think this information is important and should be included.

6)In Figure 2 the names for the OPLS-DA models are used but the OPLS-DA analysis is not used in this figure and the model names are confusing.

The model names were included as we thought that would give a more transparent overview of when the same groups had been compared. These names can easily be removed if required?

Since the OPLS-DA analysis occurs later in the manuscript I think the names should be removed from this figure.

Author Response

1. The below mentioned clarification is now added. Page 3, line 132-141,

"Peptide quality was based on total intensity in MS1 spectrum and % of total spectrum intensity contributed by identified transitions in the MS2 spectrum. The top 3 unique peptides per protein identified with highest MS1 spectrum intensity were selected. Out of these, the 1-2 peptides with the cleanest MS2 spectrum, meaning the highest % of total MS2 spectrum intensity contributed by identified peptide fragments were selected. For any protein where two peptides meeting the criterium of five good transitions (y- and b- ion signals) were selected for ordering a synthetic peptide. For any protein where only one peptide met these criterium, only one peptide was selected. Peptide identity and quality and final transition selection was subsequently verified using the synthetic peptides."

2. In Figure 2 the names for the OPLS-DA models are now removed.

Thank you for the review.

Reviewer 3 Report

Although the approach used by the authors is very interesting and their work’s value is not questionable, as declared by the authors themselves, their approach is strictly analytical and therefore I still think that the results may be more appropriated for a proteomic specialized journal and too preliminary for diagnostic purposes.

Author Response

Thank you for the kind words with regard to the performed analysis and the value of our work.